# Steering Autoregressive Music Generation with Recursive Feature Machines

**Daniel Zhao, Daniel Beaglehole, Taylor Berg-Kirkpatrick, Julian McAuley, Zachary Novack**
University of California, San Diego
{djzhao, dbeaglehole, tberg, jmcauley, znovack}@ucsd.edu

## Abstract

Controllable music generation remains a significant challenge, with existing methods often requiring model retraining or introducing audible artifacts. We introduce MusicRFM, a framework that adapts Recursive Feature Machines (RFMs) (Radhakrishnan et al., 2023) to enable fine-grained, interpretable control over frozen, pre-trained music models by directly steering their internal activations. RFMs analyze a model's internal gradients to produce interpretable "concept directions", or specific axes in the activation space that correspond to musical attributes like notes or chords. We first train lightweight RFM probes to discover these directions within MUSICGEN's hidden states; then, during inference, we inject them back into the model to guide the generation process in real-time without per-step optimization. We present advanced mechanisms for this control, including dynamic, time-varying schedules and methods for the simultaneous enforcement of multiple musical properties. Our method successfully navigates the trade-off between control and generation quality: we can increase the accuracy of generating a target musical note from 0.23 to 0.82, while text prompt adherence remains within approximately 0.02 of the unsteered baseline, demonstrating effective control with minimal impact on prompt fidelity. We release code [1] to encourage further exploration on RFMs in the music domain.

## 1 Introduction

Large autoregressive (AR) models, powered by neural audio codecs, have made remarkable strides in text-to-music (TTM) generation, producing audio with impressive fidelity and coherence (Copet et al., 2024; Yuan et al., 2025). Despite a growing body of work in conditioning TTM AR models on time-varying controls (Novack et al., 2024b;a; Wu et al., 2024; Lin et al., 2023; Koo et al., 2025), achieving precise control over fine-grained *music-theoretic* (e.g. pitch classes and chord qualities) content across time in generations remains challenging. Current approaches often focus on broad temporal controls like dynamics or polyphonic melody, and may either require intense finetuning runs or costly per-step optimization during inference to avoid large-scale training.

We argue that a more direct and principled path to controllability lies in activation-space intervention. If we can identify directions within a model's hidden states that reliably correspond to human-interpretable music-theoretic concepts, such as specific pitches, chord qualities, or tempo, we can then steer the generation along these axes, guiding the creative process without retraining the base model or altering its decoding procedure. The critical question then becomes how to discover these semantic directions in a robust and interpretable manner.

Recursive Feature Machines (RFMs) provide a powerful answer (Radhakrishnan et al., 2023; Beaglehole et al., 2025a;b). By forming an Average Gradient Outer Product (AGOP) from lightweight task probes, RFMs yield a set of orthogonal, eigenvalue-ranked directions that capture the most salient axes of variation for a given concept within a model's representation space. These directions directly represent the model's principal axes of sensitivity to specific features.

In this work, we introduce MusicRFM, the first framework that adapts RFMs for TTM generation by steering a frozen MUSICGEN-Large model directly in its activation space. Our approach is

---

[1] https://github.com/astradzhao/music-rfm

twofold: first, we train extremely lightweight, layer-wise RFM probes on the SYNTHEORY dataset (Wei et al., 2024) to extract concept-aligned directions. Then, at inference time, we inject them into the model's residual stream via forward hooks, enabling real-time, fine-grained control over the generated output. We deploy this framework on a suite of novel music-theoretic controls, controlling for diverse concepts such as the presence of specific intervallic relationships, chord qualities, and scale modes. To ensure that audio quality and fidelity is not sacrificed for steering controllability, we introduce layer-based methods that apply steering selectively across the model's 48 decoder blocks, using top-K selection or an exponential weighting scheme based on each layer's probe performance. We also show that RFMs can be used to control the presence global attributes *as a function of time*, using time-based schedules that modulate steering strength throughout the generation with functions like linear fades, sinusoidal patterns, and sparse, stochastic application. Furthermore, MusicRFM supports multi-direction steering, allowing for simultaneous or staggered enforcement of multiple attributes, such as jointly controlling notes and tempos. This comprehensive approach to control proves highly effective: our primary analysis shows that steering can increase the classification accuracy of a target note from 0.23 to over 0.82, while CLAP score for text alignment remains within ≈0.02 of the unsteered baseline—a highly favorable trade-off between control and fidelity.

In brief, MusicRFM establishes an **efficient framework** for **fine-grained, interpretable control** in **text-to-music generation**, requiring only lightweight training of RFM probes, with no finetuning or costly optimization at inference time. Its layer- and time-aware mechanisms along with support for multi-direction steering, enable controllable modulation of audio while preserving high fidelity.

## 2 RELATED WORK

Research on controllable generation spans several communities, from activation-level steering in large language models to decoding-time control methods and controllable music generation. Our work, MusicRFM, builds on and unifies these threads by adapting RFMs to the domain of music while adding new temporal and architectural control mechanisms.

### 2.1 CONTROLLABLE MUSIC AND AUDIO GENERATION

We focus particularly on text-to-music (TTM) generation that relies on neural audio codecs and autoregressive sequence models in architectures like MUSICGEN, MUSICLM, and JUKEBOX (Copet et al., 2024; Défossez et al., 2022; Agostinelli et al., 2023; Dhariwal et al., 2020; Yuan et al., 2025; Team et al., 2025). Additionally, a number of controllable TTM systems exist in the parallel diffusion domain (Novack et al., 2024b;a; Wu et al., 2024; Nistal et al., 2024a;b). Most existing controllable methods for AR focus on multi-modal controls (e.g. video) (Kim et al., 2025) or common musical controls like piano rolls (Lin et al., 2024). These approaches, while generally performant, still require reasonably compute-heavy finetuning runs and thus necessitate changing the base model; even parameter-efficient fine-tuning methods (Wu et al., 2024; Lin et al., 2024; Baker & Nistal, 2025) require 10s to 100s of GPU hours for each control added. Additionally, prior methods risk breaking the core generative capabilities of the models, especially if the finetuning data is ill-chosen.

### 2.2 ACTIVATION-LEVEL STEERING IN GENERATIVE MODELS

Beyond music, a growing body of work investigates *activation-level steering* in language models. Activation Addition (ACTADD) constructs steering vectors from paired prompts and injects them into hidden states for sentiment or style shifts, without retraining or optimization (Turner et al., 2024). Contrastive Activation Addition (CAA) extends this idea by contrasting positive/negative contexts to obtain more targeted steering directions in Llama-style models (Panickssery et al., 2024). These methods illustrate a broader trend: interpretable steering can often be achieved by modifying internal activations, rather than logits or decoding heuristics. Within music, existing approaches either focus solely on binary controls (Facchiano et al., 2025) or broad concepts like instrument presense (Koo et al., 2025), and thus it remains to be seen whether such approaches can extended to time-varying, music-theoretic control.

### 2.3 RECURSIVE FEATURE MACHINES (RFMs)

Recursive Feature Machines (RFMs) (Radhakrishnan et al., 2023) were introduced as probing methods that iteratively recondition features via AGOP matrices to uncover task-sensitive subspaces.

More recently, RFM-derived directions have been re-injected into activations for *steering* in LLMs (Beaglehole et al., 2025b). We extend this paradigm to autoregressive music generation with three innovations: (i) *layer-based control* through top-$K$ and exponential weighting across 48 layers, (ii) *time-based control* using dynamic schedules, and (iii) *multi-direction control* via simultaneous or staggered application of concept directions.

## 3 METHODS

Our overall goal is to enable fine-grained, interpretable control in AR music generation, steering towards concepts like specific notes, chord types, or slow/fast tempo. To do this, we train lightweight RFM probes to extract concept-aligned directions and re-inject them into MUSICGEN activations at inference time. This framework allows us to generate music samples that still follow text conditioning with high accuracy, while also reflecting controlled variations in targeted musical attributes.

### 3.1 BACKGROUND ON RECURSIVE FEATURE MACHINES

We first provide some more background on Recursive Feature Machines before describing our application to music generation. RFMs (Radhakrishnan et al., 2023) were originally proposed as a probing method, iteratively reconditioning features with Average Gradient Outer Product (AGOP) matrices to identify task-sensitive subspaces. Given training data $\{(x_i, y_i)\}_{i=1}^n$ and predictor $f : \mathbb{R}^d \to \mathbb{R}$, define per-sample gradients $g_i = \nabla_x f(x_i) \in \mathbb{R}^d$ and the AGOP

$$M \triangleq \frac{1}{n}\sum_{i=1}^n g_i g_i^\top \in \mathbb{R}^{d \times d}. \tag{1}$$

$M$ is PSD, with eigendecomposition $M = Q\Lambda Q^\top$. Directions $\{q_j\}$ are orthonormal, with eigenvalues $\lambda_j \geq 0$ measuring sensitivity:

$$\lambda_j = q_j^\top M q_j = \frac{1}{n}\sum_{i=1}^n (q_j^\top g_i)^2. \tag{2}$$

RFM implements *feature learning* by iterating: (i) train a base learner (kernel ridge regression as described in App. A) on features $x^{(t)}$ to obtain $f^{(t)}$, (ii) compute $M^{(t)}$ via equation 1, and (iii) update features with

$$x^{(t+1)} = T^{(t)} x^{(t)}, \quad T^{(t)} = Q^{(t)}(\Lambda^{(t)})^\alpha (Q^{(t)})^\top,$$

where $\alpha > 0$ amplifies high-sensitivity directions. Importantly, this process is *backpropagation-free*.

Recent work has extended RFMs to *steering*: injecting a concept direction $q_j$ back into hidden activations biases a frozen model toward that attribute during inference (Beaglehole et al., 2025b). In practice, steering is implemented by registering hooks on a subset of layers $S$ and adding a broadcast control vector to each residual stream:

$$h'_{t,\ell} = h_{t,\ell} + \eta_\ell(t)\, q_{\ell,j^\star}, \tag{3}$$

where $q_{\ell,j^\star} \in \mathbb{R}^{d_\ell}$ is reshaped to $(1, 1, d_\ell)$. Steering only uses the *top component* per direction.

### 3.2 MUSICRFM: RFM STEERING FOR MUSIC GENERATION

We adapt RFMs to steer MUSICGEN-large ($L=48$ decoder blocks), a Transformer over EnCodec tokens conditioned on text (Copet et al., 2024; Défossez et al., 2022). Our pipeline has three stages: (i) audio $\to$ ENCODEC codes, (ii) layerwise RFM probes that yield AGOP eigendirections, and (iii) steering applied at inference as described above.

**Synthetic Dataset for Probe Training.** SYNTHEORY (Wei et al., 2024) is a recently designed synthetic dataset made to study interpretable representations of music theory concepts in large models, divided into 7 categories: tempo, notes, chord progressions, chord types, scales, intervals, and time signatures. Compared to prior music datasets, SYNTHEORY offers clean, fine-grained supervision

of musical properties, enabling controlled experiments on model interpretability and controllability. This dataset is particularly well-suited for probing approaches, as its labeled attributes align directly with theoretical concepts that can be mapped onto latent representations. In our setting, SYNTHE-ORY allows us to train lightweight RFM probes on layerwise activations of MusicGen, yielding gradient-based directions that correspond to human-interpretable musical attributes.

**Feature extraction.** Audio clips are resampled to $32\,\mathrm{kHz}$, encoded with ENCODEC, and passed through MUSICGEN. For clip $i$ and layer $\ell$, we mean-pool over tokens, $x_{i,\ell} = \frac{1}{T}\sum_{t=1}^{T} h_{t,\ell}^{(i)} \in \mathbb{R}^{d_\ell}$, yielding clip-level vectors. Unlike last-token pooling used in text-based RFMs (Beaglehole et al., 2025b;a), mean pooling better captures temporal structure and improves probe performance.

**Probe training and steering.** For each concept $c$ and layer $\ell$, we train RFM probes for 15 iterations (fit predictor, compute AGOP, apply PSD map), keeping the probe with best validation metric (AUC for classification, MSE for regression). Binary concepts use $\{0,1\}$ labels and regression targets are z-normalized. The resulting eigendirections $q_{\ell,j}$ form interpretable axes used for steering at inference. Steering is performed by the same process described in Eq. 3. For classification tasks, we additionally train multiclass RFMs that simply replace binary labels with one-hot-encoded target vectors, predicting through softmaxing final outputs.

### 3.3 IMPROVING ROBUSTNESS IN AUDIO-DOMAIN STEERING

As we extend the existing *text*-steering framework of RFMs provided by Beaglehole et al. (2025b) to audio domain music, we introduce additional modifications to help reduce out-of-distribution behavior and improve control. In particular, given the difference between the discrete, variable-sampling rate nature of text and the continuous, fixed-sampling rate nature of audio-domain music, we found that many of the algorithmic choices made by Beaglehole et al. (2025b) were ill-suited for TTM generation. All modifications are **only applied during inference time**.

#### 3.3.1 LAYER PRUNING

Naïve steering, where we inject RFM directions uniformly across all $L{=}48$ layers at every step as is done in the original RFM paper (Beaglehole et al., 2025a), leads to noticeable degradation in audio quality and weaker alignment to text prompts. To address this, we introduce *layer pruning* strategies at inference time that prioritize informative layers and downweight noisy ones, thereby improving both perceptual fidelity and controllability (see App. C for full results).

**Top-$K$ selection.** We rank each layer $\ell \in \{1,\dots,L\}$ by its validation probe performance $\mathrm{AUC}_\ell$, then restrict steering to the top-$K$ layers.

**Exponential weighting.** Instead of hard pruning, we also apply continuous weighting across layers. For each layer $\ell$, we normalize its probe score $s_\ell$ into $\hat{s}_\ell \in [0,1]$, and define $w_\ell = w_0 \cdot \hat{s}_\ell^{1/\kappa}$ with $\kappa \in (0,1)$. This concentrates steering strength on high-performing layers, reducing unwanted artifacts and incorrect directions produced by the lower-scoring ones.

#### 3.3.2 TIME-CONTROL SCHEDULES

We modulate steering strength over time as $\eta_\ell(t) = \eta_0\, w_\ell\, \phi(t)\, \psi_p(t)$, where $\eta_0$ is a global coefficient, $w_\ell$ a layer weight, $\phi(t)$ a deterministic schedule, and $\psi_p(t)$ an optional stochastic gate.

**Deterministic schedules $\phi(t)$.** Linear/logistic *rise*, linear/exponential *decay*, and *sinusoidal* modulation let us increase or decrease a concept's influence over time (e.g., fade out a note class, ramp in a chord progression, or periodically modulate tempo). Closed-form expressions are given in App. E.

**Stochastic application $\psi_p(t)$.** At each step, apply control with probability $p$ (Bernoulli gating). Similarly to layer pruning, this method reduces over-steering and cumulative artifacts while preserving the expected bias toward the target. Ablations are in App. C.

#### 3.3.3 MULTI-DIRECTION AND STAGGERED CONTROL

We further extend MusicRFM to support *multi-direction steering*, combining multiple concept vectors $\{q_{\ell,j_m}\}_{m=1}^{M}$ in parallel. At each step we inject $h'_{t,\ell} = h_{t,\ell} + \sum_{m=1}^{M} \left[ \eta_{0,m}\, w_\ell\, \phi_m(t)\, \psi_p(t) \right] q_{\ell,j_m}$,

where each direction $m$ has its own coefficient $\eta_{0,m}$ and schedule $\phi_m(t)$. This enables both (i) *simultaneous* enforcement of multiple attributes and (ii) *staggered* control where different concepts are activated at different times. For example, one schedule may enforce tempo strongly during the opening segment, while another gradually ramps in harmonic structure later.

## 4 CLASSIFICATION RESULTS

With MusicRFM, we additionally train separate multiclass probes (different from the binary probes used to steer) to compare RFM clasification against the original probing methods used in SYN-THEORY. We see that RFMs have better or comparable performance to the 2-layer FFN probes used in the original SYNTHEORY paper across all categories. We highlight that RFMs beat the baseline probes in accuracy on scales, progressions, and intervals and in R2 score on the tempo dataset, resulting in a higher overall average score. We also find that mean-pooled RFMs outperform last-token activations. Using only the final token assumes the model compresses all relevant musical information into a single position, which is unlikely for temporally structured attributes. Mean-pooling instead aggregates information across the entire sequence, better capturing temporal patterns, especially for tempo, chord progressions, and scales.

Furthermore, we argue that FFNs do not naturally yield orthogonal, eigenvalue-ranked directions suitable for steering. In contrast, RFMs produce a PSD AGOP matrix whose eigenvectors correspond to stable, interpretable axes of sensitivity. These axes can be directly injected into the model at inference, making RFMs uniquely suited for controlled generation.

| Model | Notes | Intervals | Scales | Chords | Prog. | Time Sig. | Tempos | Avg. |
|---|---|---|---|---|---|---|---|---|
| **MusicRFM - mean pooled (ours)** | 0.850 | **0.975** | **0.956** | 0.984 | **0.943** | 0.900 | **0.985** | **0.942** |
| RFM (last token) | 0.734 | 0.743 | 0.546 | 0.866 | 0.811 | 0.771 | 0.959 | 0.776 |
| Linear Probe | 0.761 | 0.618 | 0.158 | 0.834 | 0.725 | 0.729 | 0.972 | 0.685 |
| Syntheory FFN | **0.866** | 0.972 | 0.905 | **0.989** | 0.901 | **0.905** | 0.965 | 0.929 |

Table 1: Classification results for base SYNTHEORY FFN (in Wei et al. (2024)), simple linear probes, RFMs trained on last-token activations, and **MusicRFM (ours)**. We report R2 score on the tempos dataset and accuracy on the others. We don't record performance on logistic probes as some fail to converge. Bold indicates best performing model per category.

## 5 SINGLE-DIRECTION MUSICRFM STEERING RESULTS

We report results on how well binary directions trained using MusicRFM are able to steer generations towards interpretable concepts, exploring both quantitative and subjective metrics.

### 5.1 QUANTITATIVE METRICS

We first quantify distributional shift, prompt adherence, and control accuracy of generations steered along a *single* concept direction using four metrics as a function of the control coefficient $\eta_0$: (i) **Fréchet Distance (FD)** (Gui et al., 2024) (lower is better), and (ii) **Maximum Mean Discrepancy (MMD)** (Jayasumana et al., 2024) (lower is better), (iii) **CLAP** alignment (Wu et al., 2023) using `630k-audioset-fusion-best.pt` checkpoint (higher is better), and (iv) **classification accuracy** of generated samples using the multiclass RFM probes described in Sec. 4 (higher is better).

In our setup, we compare MusicRFM steering to a simple baseline: *prompt-based conditioning*. For each concept category (e.g., notes, tempo), we append a textual hint that explicitly specifies the target attribute (e.g., *"Note: C#"* or *"Slow Tempo"*) and generate audio using MUSICGEN-LARGE without any steering. We thus compare this 3 settings: (i) a prompt-only setting, (ii) a MusicRFM-only setting, and (iii) a *combined* prompt+MusicRFM setting where the prompt conditioning and RFM directions are applied simultaneously. This allows us to disentangle what can be achieved through prompt engineering from what is uniquely enabled by RFM intervention.

For all experiments, we evaluate on a fixed evaluation set of 250 prompts sampled from the SONG-DESCRIBER dataset (Manco et al., 2023), using all 3 settings described above. For each setting,

| Category | FD ↓ | | | | MMD ↓ | | | | CLAP ↑ | | | | Probe Acc. ↑ | | | |
|---|---|---|---|---|---|---|---|---|---|---|---|---|---|---|---|---|
| | \multicolumn Control coefficient $\eta_0$ | | | | | | | | | | | | | | | |
| | 0.15 | 0.30 | 0.45 | 0.60 | 0.15 | 0.30 | 0.45 | 0.60 | 0.15 | 0.30 | 0.45 | 0.60 | 0.15 | 0.30 | 0.45 | 0.60 |
| **MusicRFM-only steering** | | | | | | | | | | | | | | | | |
| Chords (0.250) | 0.116 | 0.114 | **0.110** | 0.119 | 0.063 | 0.086 | **0.040** | 0.095 | 0.324 | **0.326** | 0.319 | **0.326** | 0.271 | 0.288 | 0.320 | **0.344** |
| Intervals (0.083) | **0.110** | 0.128 | 0.169 | 0.232 | **0.078** | 0.119 | 0.400 | 0.817 | 0.315 | **0.324** | 0.311 | 0.307 | 0.121 | 0.156 | 0.187 | **0.223** |
| Notes (0.083) | **0.113** | 0.130 | 0.138 | 0.180 | **0.052** | 0.127 | 0.217 | 0.476 | 0.315 | 0.311 | **0.318** | 0.303 | 0.231 | 0.461 | 0.684 | **0.824** |
| Scales (0.143) | **0.114** | 0.115 | 0.114 | 0.119 | **0.052** | 0.075 | 0.061 | 0.081 | 0.318 | **0.328** | 0.322 | 0.324 | 0.154 | 0.157 | 0.161 | **0.176** |
| Progs (0.053) | **0.131** | 0.142 | 0.173 | 0.207 | **0.157** | 0.233 | 0.443 | 0.650 | **0.315** | 0.309 | 0.296 | 0.297 | 0.070 | 0.079 | 0.096 | **0.114** |
| Tempos | **0.122** | 0.150 | 0.206 | 0.377 | **0.112** | 0.324 | 0.717 | 1.880 | **0.328** | 0.325 | 0.307 | 0.280 | | | — | |
| Time sigs (0.125) | **0.162** | 0.264 | 0.402 | 0.492 | **0.356** | 1.046 | 1.980 | 2.647 | **0.320** | 0.317 | 0.278 | 0.264 | 0.172 | 0.204 | 0.238 | **0.245** |
| **Prompt + RFM steering** | | | | | | | | | | | | | | | | |
| Chords (0.250) | 0.074 | **0.071** | 0.080 | 0.095 | 0.120 | **0.114** | 0.154 | 0.243 | 0.330 | 0.326 | 0.328 | **0.333** | 0.273 | 0.276 | 0.309 | **0.347** |
| Intervals (0.083) | 0.078 | **0.077** | 0.091 | 0.119 | 0.184 | **0.169** | 0.232 | 0.417 | 0.351 | **0.353** | 0.345 | 0.328 | 0.125 | 0.163 | 0.209 | **0.245** |
| Notes (0.083) | **0.108** | 0.119 | 0.133 | 0.159 | **0.438** | 0.479 | 0.563 | 0.713 | **0.343** | 0.325 | 0.321 | 0.329 | 0.657 | 0.826 | 0.921 | **0.952** |
| Scales (0.143) | 0.141 | **0.127** | 0.131 | 0.138 | 0.566 | 0.473 | **0.472** | 0.500 | **0.348** | 0.346 | 0.346 | 0.340 | 0.179 | 0.212 | 0.209 | **0.230** |
| Progs (0.053) | 0.175 | **0.170** | 0.178 | 0.186 | 0.685 | **0.670** | 0.715 | 0.758 | **0.328** | 0.314 | 0.315 | 0.298 | 0.070 | 0.085 | 0.106 | **0.129** |
| Tempos | **0.163** | 0.199 | 0.270 | 0.442 | **0.370** | 0.630 | 1.145 | 2.342 | **0.318** | 0.314 | 0.293 | 0.270 | | | — | |
| Time sigs (0.125) | **0.090** | 0.099 | 0.150 | 0.261 | 0.251 | **0.212** | 0.338 | 0.790 | **0.342** | 0.329 | 0.328 | 0.300 | 0.198 | 0.235 | 0.253 | **0.267** |
| **Prompt-only baseline** | | | | | | | | | | | | | | | | |
| Chords (0.25) | 0.069 | | | | 0.078 | | | | 0.331 | | | | 0.267 | | | |
| Intervals (0.083) | 0.082 | | | | 0.216 | | | | 0.356 | | | | 0.104 | | | |
| Notes (0.083) | 0.107 | | | | 0.414 | | | | 0.342 | | | | 0.436 | | | |
| Scales (0.143) | 0.146 | | | | 0.630 | | | | 0.344 | | | | 0.190 | | | |
| Progs (0.053) | 0.184 | | | | 0.739 | | | | 0.323 | | | | 0.065 | | | |
| Tempos | 0.087 | | | | 0.111 | | | | 0.325 | | | | — | | | |
| Time sigs (0.125) | 0.101 | | | | 0.352 | | | | 0.352 | | | | 0.139 | | | |

Table 2: Single-direction steering metrics. The top block reports RFM-only steering with stochastic application $\psi_p(t)$, $p = 0.3$ and exponential layer weighting ($w_0 = 1$, $\kappa = 0.95$). The middle block (*Prompt + RFM*) shows combined prompting and RFM steering. The bottom block (*Prompt-only*) reports baseline where only prompt is modified (independent of $\eta_0$). Parentheses denote random chance for each category. Lower is better for FD/MMD and higher is better for CLAP and Probe Accuracy (mean per-class). Ground-truth MUSICGEN-LARGE has CLAP 0.332. Probe acc is undefined for *tempos* (regression). We conduct experiments on 250 samples per class in each category.

| Steering Type | Chords | Intervals | Notes | Tempo |
|---|---|---|---|---|
| No Steering | $59.71 \pm 6.01$ | $54.75 \pm 5.52$ | $57.08 \pm 6.37$ | $55.75 \pm 7.08$ |
| Naïve RFM (ours) | $69.21 \pm 5.25$ | $62.58 \pm 5.84$ | $68.13 \pm 5.97$ | $73.33 \pm 4.35$ |
| **MusicRFM (ours, optimal)** | $73.46 \pm 4.18$ | $70.33 \pm 4.02$ | $72.88 \pm 5.67$ | $73.38 \pm 4.75$ |

Table 3: Listening test results (mean $\pm$ standard deviation) across musical attributes.

we generate 250 samples for each class in each category, and for each control coefficient $\eta_0 \in \{0.15, 0.30, 0.45, 0.60\}$ (for cases (ii) and (iii)). All results are reported on generations steered with RFM probes using stochastic application $\psi_p(t)$ with $p = 0.3$ and exponential layer weighting with $w_0 = 1$ and $\kappa = 0.95$; these are settings we found to be most optimal when creating high-quality, conceptually accurate generations. For the **tempos** category, results from each $\eta_0$ are averaged among the absolute value of the coefficient (e.g. the results from -0.15 and 0.15 are averaged into the 0.15 column). FD and MMD distributions are compared against MUSICGEN-LARGE generations produced from the original prompts without any control. We show these results in Table 2.

## 5.2 BINARY-PROBE QUANTITATIVE STEERING RESULTS

Across all categories, our quantitative metrics follow consistent trends. Distributional metrics (FD and MMD) are consistently lower at smaller control coefficients, since weak steering leaves generations closer to the reference distribution. As $\eta_0$ increases, stronger injections deviate more from ground truth and raise FD/MMD. By contrast, CLAP alignment remains essentially flat across control strengths, indicating that textual conditioning is preserved regardless of steering intensity, only with slight degradation in some categories as control coefficient increases. Probe-based classification exhibits the same monotonic behavior. Accuracy is highest for **notes**, rising sharply from 0.23 at $\eta_0$=0.15 to 0.82 at $\eta_0$=0.60, and increases monontonically for all other categories. Thus, moderate values of $\eta_0$ can balance concept control with distributional fidelity while maintaining prompt adherence. We provide additional visual graphs for the reader in App. D.

We observe that our baseline, prompt-only conditioning, except for on the notes categories, yields almost random-chance level accuracy, showing that simple textual descriptions do not provide good control. By contrast, RFM-only steering produces clear, $\eta_0$-dependent improvements. Additionally, combining prompt conditioning with RFM typically yields the strongest results, especially in cases

where prompting alone already gets accuracy to a higher level (e.g. in notes where accuracy exceeds 95% at higher $\eta_0$). These comparisons highlight that RFM activation-level steering enables forms of musical control that do not emerge from prompt engineering alone.

We note that probe accuracies should be interpreted as *relative* indicators rather than absolute ground truth. The RFM probes were trained on SYNTHEORY, a synthetic dataset with simplified musical attributes, and therefore may not generalize perfectly to natural MUSICGEN outputs. Nonetheless, their trends across $\eta_0$ provide a reliable signal that MusicRFM is correctly steering music.

| Method / $\eta_0$ | Note Dominance (%) | | | | Chord Dominance (%) | | | | Mean Event Rate (events/s) | | | | | | | |
|---|---|---|---|---|---|---|---|---|---|---|---|---|---|---|---|---|
| | 0.15 | 0.30 | 0.45 | 0.60 | 0.15 | 0.30 | 0.45 | 0.60 | -0.60 | -0.45 | -0.30 | -0.15 | 0.15 | 0.30 | 0.45 | 0.60 |
| MusicRFM | 18.50 | 34.47 | 52.50 | 66.47 | 24.40 | 28.40 | 30.50 | 35.00 | 18.66 | 20.97 | 25.07 | 26.24 | 30.48 | 30.01 | 30.88 | 31.65 |
| Prompt+RFM | 53.57 | 67.83 | 78.23 | 85.13 | 26.60 | 27.80 | 27.30 | 33.60 | 15.19 | 19.02 | 21.13 | 22.43 | 31.66 | 34.10 | 33.55 | 32.51 |
| Prompt-only | 35.97 | | | | 26.40 | | | | 25.03 (slow), 30.63 (fast) | | | | | | | |

Table 4: External evaluations across notes, chords, and tempo. Note/chord columns show target dominance (higher is better); tempo columns report mean event rate across control coefficients $\eta_0$.

### 5.3 EXTERNAL EVALUATION METRICS FOR MUSICAL CONTROL

On some categories, we introduce external evaluators that operate directly on the waveform and do not rely on our multiclass RFM probes in order to evaluate the accuracy of RFM steering. For **notes**, we compute chromagrams and label a sample as correct if the target pitch class has the highest mean energy across all classes. For **chords**, we apply an `Essentia`-based chord estimator (Bogdanov et al., 2013) and mark a sample as correct when its most frequently predicted class matches the target. Table 4 shows that RFMs work better than prompt-only injections, and accuracy increases as $\eta_0$ increases. For notes, we see an even higher accuracy increase if we combine both methods. However, for chords, we see a performance degradation, likely due to the fact that prompting alone has a low accuracy (so combining the two may push the generation in the wrong direction).

As we qualitatively found that traditional BPM detectors did a poor job of picking up on stylistic differences between generations with "fast" vs. "slow", for **tempo** we instead use a peak-weighted onset event rate (events/s) as a measure of rhythmic density (i.e. the average number of onsets per second, weighted by onset strength) using `librosa` onset detection (McFee & et al., 2023). Table 4 provides a horizontal comparison across all steering strengths. Prompt-only conditioning yields a consistent fast–slow separation, whereas RFM steering exhibits a clear monotonically increasing relationship with $\eta_0$, with a Spearman coefficient of 0.283. Combining RFM steering with prompting achieves even better results, with a Spearman coefficient of 0.433.

### 5.4 LISTENING TEST AND AUDIO SAMPLES

We provide results of a listening test, where we asked 12 participants to score 3 different audio samples for 4 control types (24 total samples, 3 control setups for 2 control examples each across 4 different control types), where they judge based on audio quality and adherence of the audio to the specified control. The 3 clips were randomly chosen base model generations (without control), naïve RFM generations, and optimal RFM generations (steering with $p = 0.3$ and exponential layer weighting with $w_0 = 1$ and $\kappa = 0.95$). Participants were randomly chosen from a departmental computer science forum at an R1 research institution, with mean age of 23.6 and mean musical experience of 9.6 years. We show mean and STD of each type of steering in Table 3. Overall, the results indicate that both naïve and MusicRFM steering substantially improve perceived control compared to the base model, with MusicRFM consistently achieving the highest ratings across all attributes. In particular, chord and interval control benefit most from RFM steering, while tempo control shows the largest relative gain over the no-steering baseline.

To the reader, we also provide representative audio samples from the listening test, illustrating single-direction control (notes), multi-direction control (notes+chords), and time-based schedules (rise/decay and crossfades). Each clip is paired with its text prompt and steering metadata ($\eta_0$, schedule), where all clips are steered with the "optimal" parameters listed above. An interactive demo of some of the clips used in our listening test is available at the project page.[2]

---

[2]https://musicrfm.github.io/controllable-music-rfm/

## 5.5 EVALUATION ON MUSICBENCH (REAL MUSIC)

To test transfer beyond synthetic data, we evaluate RFM probes on MUSICBENCH (Melechovsky et al., 2024), a real-music corpus with ground-truth tempo, notes, and keys. Using the same pipeline as in Sec. 3.2, we mean-pool MUSICGEN-large hidden states and fit layerwise RFMs (train/val/test split 70/15/15). For tempo we report normalized MSE, for classification overall accuracy. RFM probes

| $\eta_0$ | FD $\downarrow$ | MMD $\downarrow$ | CLAP $\uparrow$ | Acc. $\uparrow$ |
|---|---|---|---|---|
| 0.15 | **0.424** | **0.478** | **0.315** | 0.148 |
| 0.30 | 0.495 | 0.908 | 0.308 | 0.264 |
| 0.45 | 0.576 | 1.563 | 0.276 | 0.479 |
| 0.60 | 0.717 | 2.615 | 0.247 | **0.619** |

Table 5: RFM steering on MUSICBENCH (key).

reach 75.3% accuracy on notes and 67.5% on keys, while tempo regression proves difficult (MSE 0.862). Steering experiments (Table 5) mirror SYNTHEORY: higher $\eta_0$ increases FD/MMD and reduces CLAP, showing that moderate control preserves text adherence but aggressive coefficients destabilize generations. Overall, MusicBench confirms that real-music attributes can be steered, though sensitivity varies by concept difficulty.

## 6 MULTI-DIRECTION AND TIME-BASED STEERING RESULTS

We also evaluate MusicRFM when (i) *multiple* concept directions are injected simultaneously and (ii) when steering strength varies *over time*. We report the same quantitative metrics used in the single-direction setting and for (ii) introduce temporal analyses based on RFM probe softmax scores.

### 6.1 MULTI-DIRECTION STEERING: PAIRWISE CROSS-CATEGORY CONTROL

To test whether MusicRFM can jointly enforce *multiple musical attributes*, we examine all pairwise combinations among {**notes**, **chords**, **intervals**}. For each pair $(a, b)$, we sample a random target class from category $a$ (e.g., note $C$) and a random class from category $b$ (e.g., major chord), then generate music conditioned on both controls simultaneously.

At inference, we inject two steering directions per selected layer, one for each concept, following Sec 3.3.3. Each direction is scaled by an independent global coefficient $\eta_0 \in \{0.3, 0.6\}$. We evaluate all four cross-combinations {[0.3,0.3], [0.3,0.6], [0.6,0.3], [0.6,0.6]}, where the first value corresponds to category $a$ and the second to category $b$. For each pair, we generate $N = 100$ samples per configuration, yielding $3 \times 4 \times N = 1200$ total generations. To report results concisely, we reorganize outputs by attribute rather than by pair. For instance, all samples where *notes* were steered with coefficient 0.3—regardless of whether they were paired with chords or intervals—are averaged together. This gives us per-category summaries across all pairings, shown in Table 6.

| Concept | $\eta_0$ | FD $\downarrow$ | MMD $\downarrow$ | CLAP $\uparrow$ | Probe Acc. $\uparrow$ |
|---|---|---|---|---|---|
| Chords | 0.3 | 0.604 | 2.564 | 0.207 | 0.385 |
| Chords | 0.6 | 0.747 | 3.539 | 0.167 | 0.390 |
| Intervals | 0.3 | 0.572 | 2.351 | 0.209 | 0.298 |
| Intervals | 0.6 | 0.8861 | 4.470 | 0.134 | 0.300 |
| Notes | 0.3 | 0.566 | 2.394 | 0.205 | 0.770 |
| Notes | 0.6 | 0.927 | 4.725 | 0.133 | 0.920 |

Table 6: Multi-direction (pairwise) steering. Each cell reports the average over 200 generations.

**Findings.** We observe several trends: (i) **Probe accuracy still rises with stronger coefficients.** For notes in particular, accuracy increases from 0.770 at $\eta_0 = 0.3$ to 0.920 at $\eta_0 = 0.6$, indicating that control strength directly improves enforcement even in multi-direction cases. (ii) **Distributional metrics and CLAP scores degrade at higher strengths.** Both FD and MMD grow substantially as $\eta_0$ increases, consistent with the single-direction case, where aggressive steering pushes samples away from the reference distribution. CLAP alignment also degrades significantly. (iii) **Accuracy in multi-direction steering exceeds single-direction.** We actually observe higher probe accuracy in the multi-direction setting, which we hypothesize arises because stronger aggregate constraints reduce adherence to the text prompt (lower CLAP) and, in turn, compress the generative manifold. This yields less stylistic variance in the music, making classes easier for probes to detect.

**Interpretation.** These results highlight that multi-direction steering can indeed enforce multiple concepts, but doing so amplifies distributional drift and weakens prompt adherence. Notably, *notes* remain most controllable (large probe gains with modest $\eta_0$), while more abstract concepts like intervals yield smaller improvements. This suggests that balancing coefficients across attributes or staggering them temporally (Sec. 3.3.3), may be necessary for high-quality joint control.

## 6.2 Time-Dependent Control: Smooth Schedules

To study temporal schedules in isolation, we analyze the **notes** dataset with per-step steering strength $\eta_\ell(t) = \eta_0 \, \rho_\ell \, \phi(t)$ and track the *softmax score of the ground-truth note class* under the RFM probe as a function of time (generation steps). For the experiments in this section, we only analyze on notes because are they are the highest quality in terms of following control, and also can give us a measurable accuracy when evaluating with RFM probes.

We use per-direction coefficients $\eta_{0,m}$ and schedules $\phi_m(t) \in [0, 1]$, so $\eta_m(t) = \eta_{0,m} \, \phi_m(t)$. The schedules we ablate are exponential decay, linear decay & increase, logistic increase, and sine wave. We put formulas used in Appendix E, and record FD, MMD, and CLAP scores in Table 7.

| Schedule | FD↓ | MMD↓ | CLAP↑ |
|---|---|---|---|
| Linear increase | 0.358 | 1.917 | 0.227 |
| Linear decay | 0.321 | 1.636 | 0.257 |
| Exponential decay | 0.229 | 1.052 | 0.312 |
| Logistic increase | 0.360 | 1.999 | 0.208 |
| Sine modulation | 0.413 | 2.347 | 0.225 |

Table 7: Metrics on time-dependent controlled generations

**Correct note probability over time.** For each schedule we plot the probe softmax of the correct note over time in Figure 1a. We see that the distribution over time follows exactly what we would expect from each of the smooth scheduling functions - exponential & linear decay look like decay functions, sine is very similar to a sine wave, and logistic & linear increase show an increase in predicted probability.

**Crossfading Between Concepts.** We additionally study a controlled *cross-fade* between two notes $n_1 \to n_2$ using complementary schedules over a fixed window of 0–1500 steps: for $n_1$ we decay from $\eta_0 = 0.45$ to 0, and for $n_2$ we rise from 0 to 0.45. Formally,

$$\eta_\ell^{(n_1)}(t) = \eta_0 \cdot \phi_{\text{decay}}(t), \qquad \eta_\ell^{(n_2)}(t) = \eta_0 \cdot \phi_{\text{rise}}(t), \quad t \in [0, 1500].$$

We then log and display the RFM-probe softmax scores for both $n_1$ and $n_2$ at each timestep in Figure 1b. As expected, the first note falls in probability while the second note rises. On average over 500 randomly sampled note pairs, crossfaded generations achieve FD of $0.350$, MMD of $1.922$, and CLAP alignment of $0.250$, indicating modest distributional drift but stable prompt adherence.

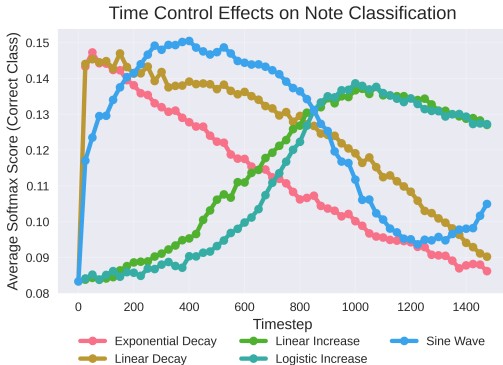

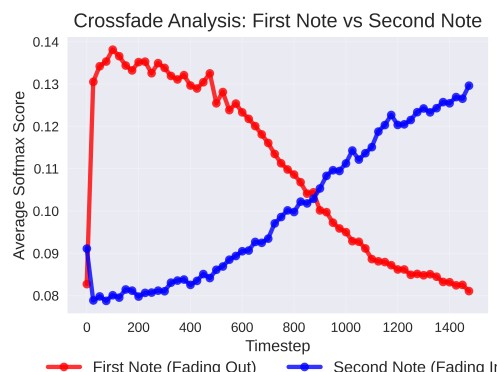

(a) Temporal softmax traces (notes). Curves show the probe probability of the ground-truth note across timesteps for different schedules (linear/exp rise/decay, log. increase, sine). We choose the probe on the best performing layer (37) as our representative probe.

(b) Two-note crossfade (softmax probabilities). The score for $n_1$ decays (red) while $n_2$ rises (blue). We again choose layer 37 as our representative probe and average over 500 samples.

Figure 1: Time-based steering analyses. (a) Probe softmax follows prescribed schedules faithfully. (b) Crossfade experiments show expected decay–rise patterns between two target notes.

## 7 LIMITATIONS & FUTURE WORK

While MusicRFM demonstrates that RFM-derived directions can steer music generation in interpretable ways, several limitations remain.

First, our probes rely on mean-pooled features, which discard temporal ordering. This limits performance on concepts with strong sequential dependencies, such as scales, chord progressions, and time signatures, where the temporal dynamics are essential for accurate classification and control. As a result, RFM probes underperform on these attributes compared to temporally local concepts like notes or chords. Future work should explore temporally aware pooling strategies (e.g., attention pooling, recurrent aggregation, convolutional pooling) or sequence-level RFMs that directly model time-evolving representations. Similarly, extending beyond the top eigenvector to incorporate multiple components could capture richer subspaces of variation, but we have not yet performed variance analyses to quantify how much information higher-order components retain.

Another promising direction is to modify the steering process itself: by selectively masking portions of the autoregressive context during inference, one could reduce model over-dependence on previously generated tokens, thus increasing the model's sensitivity to injected steering signals. Such approaches could make activation-space interventions both more controllable and also reduce the number of unwanted artifacts, particularly for longer generations.

Additionally, our experiments so far are limited to SYNTHEORY-based, symbolic music-theoretic concepts such as notes, chords, and tempo. Future work could extend MusicRFM to attributes more directly tied to perceptual or production-level qualities, including instrument identity, timbre, or articulation style. While we perform preliminary analysis on MUSICBENCH, extended RFM training and steering on real-music-based datasets remains an open direction. These studies would connect RFM steering more directly to interpretability in real-world generation tasks.

Finally, our experiments target MUSICGEN-large, but other large audio models open complementary directions for RFM steering. OpenAI's JUKEBOX (Dhariwal et al., 2020) uses multi-scale VQ-VAE codes and hierarchical AR decoders, while Google's recent MAGENTA-RT (Team et al., 2025) framework supports *real-time* audio generation. Applying RFMs in these contexts would require adapting probe extraction to multi-level codebooks (for Jukebox) and to low-latency streaming architectures (for Magenta-RT). In particular, real-time models highlight the possibility of **real-time steering**: dynamically injecting directions during live playback, enabling interactive control (i.e. live DJ-ing). Extending MusicRFM into these setups could bridge interpretability with performance-critical generative applications such as interactive music tools and live performance systems.

## 8 CONCLUSION

We presented *MusicRFM*, a framework that leverages RFM-derived, eigenvalue-ranked directions to steer a frozen MUSICGEN-large model directly in activation space. By combining concept-aligned directions with layer-aware weighting and time-dependent schedules, MusicRFM enables fine-grained, interpretable control over attributes such as notes, chords, and tempo without modifying the base model or relying on per-step optimization.

Across synthetic and real-music settings, we observed consistent trade-offs governed by the control coefficient $\eta_0$: moderate steering improves alignment to targeted concepts with limited distributional drift (FD/MMD) and minimal degradation in prompt adherence (CLAP), while aggressive steering yields stronger control at the cost of artifacts. Notes are the most reliably controllable, multi-direction steering is feasible but amplifies drift, and simple schedules (e.g., decay/rise) support intuitive manipulations like crossfades. Time-based control is accurate and true-to-schedule in terms of evaluating on softmax probability of classes. Layer pruning and stochastic (Bernoulli) application help stabilize generations by limiting cumulative bias.

By enhancing fine-grained controllability, this line of research can significantly broaden the practical applications of generative models. In the long term, improving the steerability and interpretability of generative models will expand their usefulness in domains like music production and game audio.

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

## A OVERVIEW OF KERNEL RIDGE REGRESSION

Kernel ridge regression (KRR) is the base model with which we apply the RFM procedure for iterative feature learning via the AGOP. We briefly explain the KRR model. Let $X \in \mathbb{R}^{n \times d}$ denote training samples with $x^{(i)^T}$ denoting the sample in the $i^{\text{th}}$ row of $X$ for $i \in [n]$ and $y \in \mathbb{R}^{n \times c}$ denote training labels, where $c$ is the number of output channels (e.g. one-hot encoded classes for $c > 2$ classes). Let $K : \mathbb{R}^d \times \mathbb{R}^d \to \mathbb{R}$ denote a kernel function (a positive semi-definite, symmetric function), such as the Gaussian/RBF kernel ($K(x, z) = \exp(-\|x - z\|_2^2)/L^2)$, or the Laplace kernel ($K(x, z) = \exp(-\|x - z\|_2)/L$) used in this work. Given a ridge regularization parameter $\lambda \geq 0$, KRR solved on the data $(X, y)$ gives a predictor, $\hat{f} : \mathbb{R}^d \to \mathbb{R}^c$, of the form:

$$\hat{f}(x) = K(x, X)\alpha \,, \tag{4}$$

where $\alpha$ is the solution to the following linear system:

$$(K(X, X) + \lambda I)\alpha = y \,. \tag{5}$$

Here the notation $K(x, X) \in \mathbb{R}^{1 \times n}$ is the $n$-dimensional row vector with $K(x, X)_i = K(x, x^{(i)})$ and $K(X, X) \in \mathbb{R}^{n \times n}$ denotes the kernel matrix of pair-wise kernel evaluations $K(X, X)_{ij} = K(x^{(i)}, x^{(j)})$. The advantage of kernel functions in the context of this work is that the predictor admits a closed form solution, which can be robustly computed and generally has fast training times for datasets under 70k samples.

## B TUNING PROCEDURE FOR RFM PROBING

We use 70/15/15 train/valid/test split on RFM training, 15 RFM iterations, and mean pooling over all timesteps. For multiclass training of simple progressions, we use 700 examples per class (there are 1100 per class in dataset, but we cannot fit them given our A6000 GPU memory size. However, we note that even without all training data, we still get significantly better accuracy than baseline in this category). For all other classes, we use the entire dataset for our training & validation. We use 100 random choices of hyperparameters listed below for layer-wise probes and 300 for aggregation. We maximize on AUC for layer-wise probes and accuracy for aggregation.

When tuning the number of components calculated with our RFM probes, we tried a lower number of components (2-10) for categories with less data points and less perceived complexity (e.g. notes, time signatures). For categories with larger dataset size and higher perceived complexity (e.g. simple progressions, scales), we choose number of components ranging from 8 to 24.

| Hyperparameter | Layer-wise | Aggregation model |
|---|---|---|
| Bandwidth (L) | $\log \mathcal{U}(1, 100)$ | $\log \mathcal{U}(1, 100)$ |
| Center gradients | {False, True} | {False, True} |
| Exponent $q$ | $\mathcal{U}(0.7, 1.4)$ | $\mathcal{U}(0.7, 1.4)$ |
| Kernel Type | $K_{2,q}$ | $K_{p,q}$ |
| $p$ (when kernel type is $K_{p,q}$) | – | $\mathcal{U}(q, 2)$ |
| Regularization | $\log \mathcal{U}(10^{-5}, 10)$ | $\log \mathcal{U}(10^{-5}, 10)$ |

Table 8: Search spaces for MusicRFM on individual layers and for the aggregation model.

Note we tune over a more general class of kernels $K_{p,q}(x, x') = \exp(-\|x - x'\|_p^q/L^q)$ (indicated by the kernel type hyperparameter) for the aggregation model, which has been shown to improve the performance of RFM on tabular datasets (Beaglehole et al., 2025a). We also tune over whether to center the gradients in each iteration of RFM, which can help de-noise the gradients in high-dimensional settings (Beaglehole et al., 2025b). Gradient centering modifies the AGOP computation to give the following centered M matrix in the RFM iteration, where $\bar{g} = \frac{1}{n} \sum_{i=1}^{n} g_i$:

$$M^{(t)} = \frac{1}{n} \sum_{i=1}^{n} (g_i - \bar{g})(g_i - \bar{g})^\top \,. \tag{6}$$

## C  STEERING ABLATIONS

For generation, we ablate two steering knobs that most strongly impact generation quality and concept alignment: (i) the *effective number of layers* contributing control via both a flat top-$k$ value and an exponential, score-weighted layer scheme ("layer pruning"), and (ii) a *per-timestep injection probability* $p$ that sparsifies when control is applied.

### C.1  SETUP AND METRICS

We follow Sec. 3.2 and inject layerwise RFM directions into the residual stream with strength

$$\eta_\ell(t) \;=\; \eta_0 \, \rho_\ell \, \phi(t) \, \psi_p(t),$$

where for ablations we set $\phi(t) \equiv 1$ and vary layer weighting and $p$.

### C.2  ABLATING LAYER PRUNING

We study three strategies that control how many (and how strongly) layers contribute to steering: (i) *exponential* score-weighted steering, (ii) a simple *linear* score-weighted scheme, and (iii) hard *top-$K$* selection. We show results in Table 10 and Table 9.

**Continuous weighting (Linear vs. Exponential).**  Given base scale $w_0$, we instantiate the per-layer weight $\rho_\ell$ in $\eta_\ell(t) = \eta_0 \, \rho_\ell \, \phi(t)$ using either:

$$\textbf{Linear:} \quad w_\ell^{\text{lin}} \;=\; w_0 \, \hat{s}_\ell, \qquad\qquad \textbf{Exponential:} \quad w_\ell^{\text{exp}} \;=\; w_0 \, \hat{s}_\ell^{\,1/\kappa},$$

where $\kappa \in (0, 1)$ is a *decay rate* (smaller $\kappa$ increases contrast, concentrating weight on high-scoring layers). Linear is the minimal "from 1 to 0" mapping; exponential recovers linear as $\kappa \to 1$ and becomes more selective as $\kappa \downarrow$.

**Discrete selection (Top-$K$).**  We also ablate a hard selection mask $m_\ell^{(K)} \in \{0, 1\}$ over the top-$K$ layers by $\hat{s}_\ell$:

$$m_\ell^{(K)} \;=\; \mathbb{1}[\ell \in \text{TopK}(\hat{s}, K)], \qquad w_\ell^{\text{top-}K} \;=\; w_0 \, m_\ell^{(K)}.$$

We sweep $K \in \{4, 8, 12, 16, 24, 32, 48\}$, with $K{=}48$ meaning all layers.

| Scheme | Hyperparams | FD ↓ | MMD ↓ | CLAP ↑ | Classification Acc. ↑ |
|---|---|---|---|---|---|
| Linear | $w_\ell = w_0 \hat{s}_\ell$ | 0.482 | 2.701 | 0.166 | 0.959 |
| Exponential | $\kappa = 0.98$ | 0.487 | 2.710 | 0.186 | 0.954 |
| Exponential (ours) | $\kappa = 0.95$ | 0.465 | 2.575 | 0.194 | 0.961 |
| Exponential | $\kappa = 0.92$ | 0.483 | 2.687 | 0.175 | 0.954 |
| Uniform (naive) | – | 0.599 | 3.44 | 0.155 | 0.964 |

Table 9: Layer *weighting* ablation (continuous schemes). Exponential decay $\kappa$ interpolates between flat ($\kappa \to 1$) and highly concentrated ($\kappa \to 0$). Linear maps the best layer to $w_0$ and the worst to 0.

| Top-$K$ | FD ↓ | MMD ↓ | CLAP ↑ | Classification Acc. ↑ |
|---|---|---|---|---|
| $K = 4$ | 0.109 | 0.192 | 0.309 | 0.398 |
| $K = 8$ | 0.157 | 0.448 | 0.291 | 0.678 |
| $K = 12$ | 0.225 | 0.919 | 0.263 | 0.882 |
| $K = 16$ | 0.347 | 1.781 | 0.225 | 0.941 |
| $K = 24$ | 0.555 | 3.218 | 0.158 | 0.967 |
| $K = 32$ | 0.586 | 3.395 | 0.158 | 0.958 |
| $K = 48$ (naive) | 0.599 | 3.44 | 0.155 | 0.964 |

Table 10: Layer *selection* ablation (top-$K$ hard pruning). $K$ controls the effective number of controlled layers.

### C.3 ABLATING INJECTION PROBABILITY $p$

At each generation step $t$, we sample a gate $b_t \sim \text{Bernoulli}(p)$ and apply:

$$h'_{t,\ell} \;=\; h_{t,\ell} \;+\; b_t\, \eta_\ell(t)\, q_\ell,$$

so control fires stochastically with probability $p$. We show results in Table 11.

| $p$ | FD $\downarrow$ | MMD $\downarrow$ | CLAP $\uparrow$ | Classification Acc. $\uparrow$ |
|---|---|---|---|---|
| 0.15 | 0.108 | 0.163 | 0.348 | 0.348 |
| 0.30 (ours) | 0.118 | 0.272 | 0.306 | 0.697 |
| 0.45 | 0.197 | 0.769 | 0.287 | 0.884 |
| 0.6 | 0.281 | 1.343 | 0.265 | 0.931 |
| 0.75 | 0.399 | 2.145 | 0.207 | 0.961 |
| 0.9 | 0.510 | 2.853 | 0.172 | 0.962 |
| 1.0 (naive) | 0.599 | 3.44 | 0.155 | 0.964 |

Table 11: Injection probability ablation. Lower $p$ reduces artifacts but may weaken alignment; higher $p$ increases control strength but risks over-steering.

## D SINGLE DIRECTION METRIC GRAPHS

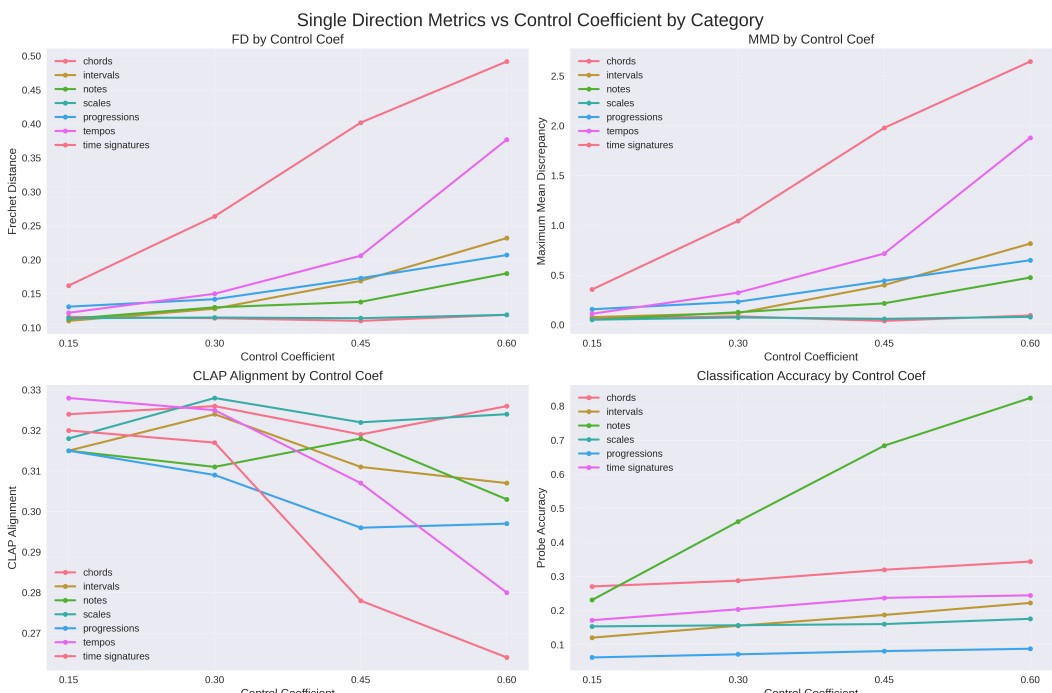

Figure 2: Single-direction steering metrics as a function of control coefficient $\eta_0$. Top Left: Fréchet Distance (FD; $\downarrow$) increases with stronger control. Top Right: Maximum Mean Discrepancy (MMD; $\downarrow$) shows a similar trend. Bottom Left: CLAP alignment ($\uparrow$) to the text prompt remains relatively stable for most categories. Bottom Right: Probe accuracy shows that, despite poor performance on generated data, there is an upwards trend in accuracy as we increase the control coef $\eta_0$.

# E    Control schedules used for time control ablations on note classification

$$\phi_{\text{lin}\uparrow}(t) = \min\left(\max(\tfrac{t}{1500}, 0), 1\right), \quad \phi_{\text{lin}\downarrow}(t) = 1 - \min\left(\max(\tfrac{t}{1500}, 0), 1\right),$$

$$\phi_{\text{exp}\downarrow}(t) = \lambda^t, \quad (\lambda = 0.998), \qquad \phi_{\text{log}\uparrow}(t) = \tfrac{1}{1+\exp(-(t-750)/200)},$$

$$\phi_{\sin}(t) = \tfrac{1}{2}\left(1 + \sin(2\pi t/1500)\right).$$

# F    RFM Steering Pseudocode

---
**Algorithm 1** MusicRFM steering
---
1: **Input:** Directions $\{q_{\ell,c}\}$; control scale $\eta_0$; layer weights $w_\ell$; schedule $\phi(t)$; gate probability $p$, total timesteps $T$.
2: **Output:** Generated sequence $(y_1, \ldots, y_T)$.
3: $\boldsymbol{y} = \{BOS\}$
4: **for** $t = 1$ to $T$ **do**
5: $\quad h_{t,0} = \text{TokenEmbed}\,(\boldsymbol{y})$
6: $\quad$ **for** $\ell = 1$ to $L$ **do**
7: $\qquad h_{t,\ell} = \text{TransformerBlock}_\ell(h_{t,\ell-1})$
8: $\qquad$ **if** $w_\ell > 0$ **then**
9: $\qquad\quad$ **if** $\text{Bernoulli}(p) = 1$ **then**
10: $\qquad\qquad \eta_\ell(t) \leftarrow \eta_0\, w_\ell\, \phi(t)$
11: $\qquad\qquad h_{t,\ell} = h_{t,\ell} + \eta_\ell(t)\, q_{\ell,c}$
12: $\qquad\quad$ **end if**
13: $\qquad$ **end if**
14: $\quad$ **end for**
15: $\quad \boldsymbol{y} \leftarrow \boldsymbol{y} \bigoplus \text{Sample}(h_{t,L})$
16: **end for**
17: **return** $\boldsymbol{y}$
---

