# OpenReview forum: "Steering Autoregressive Music Generation with Recursive Feature Machines"
_ICLR.cc/2026/Conference — ICLR 2026 Poster_

### Official Review · Reviewer_6iX2 · 2025-10-30

**Soundness:** 3
**Presentation:** 3
**Contribution:** 2
**Rating:** 6
**Confidence:** 5

**Summary:**

This paper introduces MusicRFM, a framework for activation-level steering of pre-trained autoregressive music models. Building on RFMs, the authors identify interpretable “concept directions” in MusicGen’s hidden states that correlate with musical attributes such as notes, chords, or tempo. These directions are injected into the model’s residual stream during inference, allowing fine-grained control without retraining or step-wise optimization. Experiments on synthetic datasets (SYNTHEORY) and real-music benchmarks (MUSICBENCH) evaluate classification accuracy, FAD/MMD/CLAP metrics, and small-scale listening tests.

**Strengths:**

The proposed approach enables interpretable, fine-grained control over musical attributes such as notes, chords, and tempo without retraining or per-step optimization. The introduction of layer-aware steering, time-varying schedules, and multi-directional control makes the framework flexible and musically relevant. Empirical results demonstrate effective controllability with minimal loss of fidelity, showing a clear advancement toward interpretable and lightweight control in music generation systems.

**Weaknesses:**

The paper lacks essential experimental details, including the datasets, prompts, and number of samples used for evaluation. While the experiments appear to partially rely on the SynTheory dataset, this is never explicitly stated or described in detail, making it difficult to assess reproducibility. In addition, the absence of a direct baseline would be the critical issue of the paper. Most naively, authors can compare the proposed steering approach to a simple text-conditioning (e.g., simply adding “fast music” to the input prompt) limits the clarity of the paper’s contribution. Such a comparison is crucial to demonstrate the unique benefits of activation-space control over standard prompt-based methods.

**missing references**

- citation for metrics? (FD, MMD, CLAP) it’s not mentioned of which CLAP model was used for evaluation

**minor**

- Table 1: correct the caption (”We train using 7 We report…”) and place the proposed method at the most bottom row
- reporting FD and MMD together is a bit redundant
- Table 2 caption: “higher better for” → “higher is better for”
- line 315: “were randomly chosen base model …” → “were randomly chosen from base model ...”
- couldn’t find listening examples for time-based schedules from the demo page upon time of reviewing

**Questions:**

- Table 1: what is the implication of performance difference between proposed and RFM (last token)?
- Table 2: mae for Tempos?
- Table 2: Classification results were very high from Table 1. Doesn’t this mean it’s a bit reliable to trust the metric?
    1. If the authors claim it’s not reliable for real music (according to Table 2 caption), then correlation between human evaluation should be included to observe how different it is and find the best way for evaluation (e.g., models specific for each downstream task)
    2. If this Probe Acc is reliable, then the successful controllability is below chance level for all tasks except for Notes. Is this acceptable? Especially since there’re no other baselines to compare.
- details of participants of listening test? and 3 questionnaire seems too small
- what’s the benefit compared to diffusion-based controls
- so what is the best $\eta_0$? and any way to automate this for each steering direction?

---

> ### Author Response · Authors · 2025-11-21
> **Rebuttal #1**
>
> We thank the reviewer for their insightful comments, and are glad they recommend acceptance. Below, we address the concerns brought up in the review:
>
>
> >**citation for metrics? (FD, MMD, CLAP) it’s not mentioned of which CLAP model was used for evaluation**
>
>
> We thank the reviewer for bringing to our attention the lack of citations on these metrics. We have now added proper citations to the paper and included the exact CLAP model used.
>
> >**reporting FD and MMD together is a bit redundant**
>
> We understand the reviewers point of view, and chose to report FD and MMD together following a number of recent papers [1,2], as well as the documented occurrence that FD and MMD may capture different things [3,4].
>
> [1] Novack, Zachary, et al. "Presto! distilling steps and layers for accelerating music generation." arXiv preprint arXiv:2410.05167 (2024).
>
> [2] Nistal, Javier, et al. "Diff-a-riff: Musical accompaniment co-creation via latent diffusion models." arXiv preprint arXiv:2406.08384 (2024).
>
> [3] Huang, Yichen, et al. "Aligning Text-to-Music Evaluation with Human Preferences." arXiv preprint arXiv:2503.16669 (2025).
>
> [4] Grötschla, Florian, et al. "Benchmarking Music Generation Models and Metrics via Human Preference Studies." ICASSP 2025-2025 IEEE International Conference on Acoustics, Speech and Signal Processing (ICASSP). IEEE, 2025.
>
>
> >**The paper lacks essential experimental details...In addition, the absence of a direct baseline would be the critical issue of the paper. Most naively, authors can compare the proposed steering approach to a simple text-conditioning (e.g., simply adding “fast music” to the input prompt) limits the clarity of the paper’s contribution.**
>
> We thank the reviewer for raising these concerns. Regarding prompts and sample counts: each concept category uses a fixed set of 250 prompts from Song Describer. We also have added explicit training details are included in the new version (Section 5.1) for full reproducibility.
>
> Most importantly, we now include the prompt-only conditioning baseline that the reviewer suggested, where we simply append the target attribute to the input prompt (e.g., “Note: C♯”, “Major chord”, “Fast tempo”). We also combine this with our RFMs, to see whether any beneficial aspects of prompt-based conditioning can work in synchronicity with our RFMs. Across all attributes, prompt-only control is weak (e.g., 36% note accuracy, 26% chord accuracy), while RFM steering provides much stronger controllability (66% and 35%, respectively), and the combined prompt+RFM setting performs best overall in cases where prompting exhibits improvement over random chance. This directly demonstrates that activation-space steering offers capabilities that prompt engineering alone does not provide.
> We believe your suggestion has significantly improved the quality of our experimental results, as prompting (when useful) can be combined with RFMs to significantly improve all metrics. We hope these clarifications and additions address the reviewer’s concerns.
>
> >**couldn’t find listening examples for time-based schedules from the demo page upon time of reviewing**
>
> We thank the author for mentioning this and have updated the website to include many more samples, including ones with time-based scheduling and multi-direction steering.
>
> >**Table 2: Classification results were very high from Table 1. Doesn’t this mean it’s a bit reliable to trust the metric?**
>
> We thank the reviewer for bringing this up and would like to clarify some things. We do think that there is some mismatch with the probes trained on SynTheory and evaluating on generations (as syntheory outputs are synthetic and prototypically simple), and have such added further external control evaluation to better consolidate our findings. Specifically, we added a chromagram-based method for notes, an Essentia-based chord-quality estimator for chords, and an onset-density estimator for tempo. We hope that results from these evaluators can provide more perspective on the ability to use RFMs for musical control.
> However, we’d also like to note that “random chance” here is not 50%, but rather $ 1 / C$ where $C$ is the number of classes for the particular control. We have added this number to Table 2 in parentheses, and show that across all categories, even at the lowest control coefficients we achieve better than random performance.

---

> > ### Author Response · Authors · 2025-11-21
> > **Rebuttal #2**
> >
> > >**Table 1: what is the implication of performance difference between proposed and RFM (last token)?**
> >
> > We thank the reviewer for giving us the opportunity to clarify this. We believe that mean-pooling RFMs provide better results because relying on final token representations assumes that the base model is able to compress all relevant information into the final position, which may not be true for most temporally based musical attributes. Instead, mean-pooling directly pools information across the entire sequence to provide better context to the RFM on what internal representations and gradients make the most difference across the entire sequence. This is especially important for classes like tempo, time signature, scales and chord progression, which need temporal information to be classified correctly. We additionally add this information to Section 4 in our paper.
> >
> > >**Table 2: mae for Tempos?**
> >
> > We thank the reviewer for pointing this out. Because our RFM probes for tempo were trained on synthetic SynTheory data, where each tempo class is represented by simple drum patterns at fixed BPMs, a direct MAE on continuous BPM is not meaningful, as our probe was never trained to output real-music based tempo estimates. Instead, we find that what the probe learns is essentially event density (how frequently onsets occur, and what is encoded in syntheory).
> >
> > For this reason, we report the weighted event density (i.e. events / s, where each event is weighted by its overall magnitude) using librosa and the Spearman rank coefficient across control coefficients, which capture the overall rhythmic density and how well such density correlates across increasing control coefficients respectively. Here we see that results are quite good, with event-rate increasing as the control coefficient goes from negative to positive, and a spearman of 0.283 for pure RFMs 0.433 for RFMs+Prompts.
> >
> > >**details of participants of listening test? and 3 questionnaire seems too small**
> >
> > We thank the reviewer for bringing this up, and have since amended the paper to include these details. Participants were randomly selected from a public computer science forum at an R1 research institution, with mean age of 23.6 and mean musical experience of 9.6 years. Samples were randomly generated from each of the three setups, and all participants rated all 24 samples. We also realized that we did not fully explain this section well in our original draft and have since amended it. In particular, the subjects were given sets of 3 samples (MusicRFM, baseline RFM, and no control) for 2 prompts across 4 different control categories, for a total of 24 samples.
> >
> > >**what’s the benefit compared to diffusion-based controls**
> >
> > We thank the reviewer for giving the opportunity to address this. In general, we posit MusicRFM as sitting orthogonal to diffusion-based controls, as they fundamentally deal with distinct model classes that have their own benefits and tradeoffs. Most clearly, MusicRFMs offer the first lightweight control for AR codec models, and while diffusion models have their own suite of inference-time controls (guidance, optimization), these are unusable for a growing proportion of open-source AR models (MusicGen, YuE, Magenta-RT, SongGen).
> >
> > In comparison to inference-time controls for diffusion models (as training-based ones are mostly equivalent to training-based AR methods), RFMs have the added benefit of incurring 0 additional inference cost (at the cost of the short offline training of RFMs), while both guidance and optimization methods for diffusion incur substantial slowdowns. Additionally, in theory there is no reason why RFMs couldn’t also work for diffusion models (as RFMs only require a transformer-like architecture, which many modern diffusion models have, not any sort of causal relationship), and such RFMs may be expandable into diffusion as well, which we leave for future work.
> >
> > >**so what is the best $\eta_0$ ? and any way to automate this for each steering direction?**
> >
> > We thank the reviewer for this concern. The control coefficient itself is not something that can be optimized automatically - we believe it is much more dependent on user preference. Because it basically determines the tradeoff between prompt fidelity and controllability, we see it more as a user-determined value rather than an objective metric. If the user wants to preserve the original prompt more closely, they may want to use smaller coefficients, and if they want stronger control, they would use larger coefficients. Because different applications value these tradeoffs differently, we treat the control coefficient as an intentional user-facing knob rather than a hyperparameter to optimize.

---

### Official Review · Reviewer_h1Zj · 2025-11-01

**Soundness:** 3
**Presentation:** 3
**Contribution:** 3
**Rating:** 8
**Confidence:** 5

**Summary:**

This paper introduces MusicRFM, a framework that adapts Recursive Feature Machines (RFMs) to enable fine-grained control over frozen autoregressive music generation models like MUSICGEN. By training lightweight RFM probes on the SYNTHEORY dataset, the authors extract interpretable "concept directions" corresponding to musical attributes (e.g., notes, chords, tempos). These directions are injected into the model's activations during inference to steer generation without retraining or per-step optimization. Key innovations include layer-based pruning (top-K and exponential weighting), time-varying schedules (e.g., linear fades, sinusoidal modulation), and multi-direction steering for simultaneous control of multiple attributes. Experiments demonstrate improved control accuracy (e.g., note classification from 0.23 to 0.82) with minimal impact on text prompt fidelity (CLAP score within ~0.02 of baseline), supported by quantitative metrics (FD, MMD, CLAP) and a small listening test.

**Strengths:**

This is a solid paper that explores the application of Recursive Feature Machines (RFMs) to music generation, specifically for steering codec-based autoregressive models like MusicGen.

The work addresses an important challenge in controllable music generation by enabling fine-grained, interpretable control over musical attributes such as notes, chords, intervals, scales, progressions, tempos, and time signatures, without requiring model retraining or heavy optimization.

The use of the SYNTHEORY dataset for probe training is well-motivated, as it provides clean, music-theoretic labels.

The proposed extensions—layer pruning strategies (top-K and exponential weighting), dynamic time schedules, and multi-direction steering—are novel and practical, allowing for more robust and flexible control.

Experiments are comprehensive, including classification results showing RFMs outperforming baselines (e.g., average score 0.942 vs. 0.929 for SYNTHEORY FFNs), quantitative metrics on steering trade-offs (FD, MMD, CLAP, probe accuracy), and a listening test demonstrating perceptual improvements.

Overall, the paper makes a meaningful contribution to activation-level steering in generative audio models, with potential for broader applicability.

**Weaknesses:**

First, the related work section underestimates prior methods by claiming they require "intense finetuning runs," when many actually use parameter-efficient fine-tuning (PEFT) techniques, which are computationally lighter than full finetuning (though still more than RFMs). This portrayal lacks objectivity and overlooks the fact that some methods already achieve control over notes and chords without architectural gaps.

Additionally, relevant papers are missing, such as Zhu et al. (2025) on efficient fine-grained guidance for diffusion-based symbolic music generation via inference-time control, and Zhang et al. (2024) on zero-shot text-to-music editing with diffusion models, which also focuses on inference-time interventions. Second, although the writing is clear overall, the methods section can be dense and challenging to follow, particularly the details on RFM adaptations, layer pruning, and schedule formulations; more explanatory text or examples would help.

Third, objective experiments show that increasing steering strength degrades FD and MMD scores, which seem to imply a degradation in musicality as well—objective metrics like FD and MMD are designed to capture distributional shifts that often correlate with perceptual quality, suggesting that stronger steering may introduce artifacts or incoherence that harm the overall musical experience. This is my biggest concern, as it raises questions about the method's ability to maintain high-fidelity generations under aggressive control, yet the paper lacks subjective evaluations of fidelity and musicality beyond the small listening test—relying on proxies like CLAP may not fully capture these aspects, potentially limiting the method's practical value.

Finally, the listening test protocol uses only 12 participants, which is insufficient for robust conclusions, and omits details on participant selection criteria, demographics, or distribution.

[1] Zhu, T., Liu, H., Wang, Z., Jiang, Z., & Zheng, Z. Efficient Fine-Grained Guidance for Diffusion Model Based Symbolic Music Generation. In Forty-second International Conference on Machine Learning.

[2] Zhang, Y., Ikemiya, Y., Xia, G., Murata, N., Martínez-Ramírez, M. A., Liao, W. H., ... & Dixon, S. (2024). Musicmagus: Zero-shot text-to-music editing via diffusion models. arXiv preprint arXiv:2402.06178.

**Questions:**

1. Could the authors address the underestimation of related work? For instance, how does MusicRFM compare directly to PEFT-based methods in terms of computational cost and control granularity? Also, why were papers like Zhu et al. (2025) and Zhang et al. (2024) not discussed, given their focus on inference-time control in music generation?

2. To improve readability, could the authors suggest additions to the methods section, such as pseudocode for the steering injection process or a simple worked example of a time schedule (e.g., linear rise) applied to a generation?

3. Regarding the trade-off between control strength and metrics like FD/MMD, do the authors have plans for larger-scale subjective evaluations of audio fidelity and musicality? How might this affect the method's usability in real-world applications, and are there mitigation strategies?

4. For the listening test, could the authors provide more details on the protocol? Specifically, what were the participant demographics, expertise levels (e.g., musicians vs. general listeners), and how were samples randomized?

**Details Of Ethics Concerns:**

This paper comes with a subjective experiments, usually needs an ethics approval.

---

> ### Author Response · Authors · 2025-11-21
> **Rebuttal #1**
>
> We thank the reviewer for their insightful comments, and are glad they recommend acceptance. Below, we address the concerns brought up in the review:
>
>
> >**Could the authors address the underestimation of related work? For instance, how does MusicRFM compare directly to PEFT-based methods in terms of computational cost and control granularity? Also, why were papers like Zhu et al. (2025) and Zhang et al. (2024) not discussed, given their focus on inference-time control in music generation?**
>
> We thank the reviewer for pointing this out, and have updated our related work to provide a more unbiased discussion of fine-tuning methods. In general, while PEFT methods are less expensive than full fine-tuning and from-scratch training, their benefits are more tied to reducing the memory overhead of training rather than the overall speed (though there are reasonable wall-clock speedups). In fact, many PEFT methods still require reasonable heavy fine-tuning setups, with statistics from published works including over 500 GPU hours for Music ControlNet, using 4 RTX8000s (48GB VRAM each) for CocoMulla, and 2 days of training per control for LiLAC. In contrast, each RFM takes less than a minute to train on a single GPU. Additionally, it is unclear whether any method for PEFT would naturally work on synthetic data like Syntheory, whereas RFMs can be successfully trained on synthetic data while still steering real audio. Regarding the aforementioned papers, these were not initially discussed as these are both on *diffusion* models, rather than AR codec-based models, and are both reasonably distinct from the present task setting (Zhu et al. deals with symbolic rather than audio domain music, while Zhang et al. deals with music editing rather than control). However, we agree with the author that their inclusion would make the related works more complete, and have such amended the section.
>
> >**To improve readability, could the authors suggest additions to the methods section, such as pseudocode for the steering injection process or a simple worked example of a time schedule (e.g., linear rise) applied to a generation?**
>
> We thank the reviewer for this suggestion. While we do include explicit details of the time schedules used in Appendix E, we have expanded this section to include a formal pseudocode algorithm describing the steering injection process (Appendix F).
>
> >**Regarding the trade-off between control strength and metrics like FD/MMD, do the authors have plans for larger-scale subjective evaluations of audio fidelity and musicality? How might this affect the method's usability in real-world applications, and are there mitigation strategies?**
>
> The reviewer makes a valuable point regarding wider scale evaluation. While we do not intend to perform large scale subjective evaluations beyond our initial subjective listening test given the practical constraints of the 2-week, we have added additional external evaluation for the chords, notes, and tempo categories. We hope that this will give more concrete evidence that RFMs can control music generation accurately.
>
> Regarding the trade-off, we note that the balance between control and fidelity is well documented in a wide array of features of generative modeling (e.g. classifier-free guidance increasing control at the cost of oversaturation, LLM temperature increasing quality at the expense of diversity), and thus find this trade-off to be amenable in the present manuscript, and intend to improve upon this in future work.
>
> >**For the listening test, could the authors provide more details on the protocol? Specifically, what were the participant demographics, expertise levels (e.g., musicians vs. general listeners), and how were samples randomized?**
>
> We thank the reviewer for bringing this up, and have since amended the paper to include these details. Participants were randomly selected from a public computer science forum at an R1 research institution, with mean age of 23.6 and mean musical experience of 9.6 years. Samples were randomly generated from each of the three setups, and all participants rated all 24 samples.
>
> >**This paper comes with a subjective experiments, usually needs an ethics approval.**
>
> We wanted to address the purported concerns from the reviewer over the ethics review. In our work, we have taken academic ethics very seriously, received clearance from academic supervisors, and ensured that there was no way in which the subjective listening test could cause harm to participants (in practice, participants simply listened to 24 music samples with benign text captions, and were allowed to set their own volume levels). If the reviewer has specific concerns regarding our subjective listening test, we would be more than welcome to address them during the rebuttal period.

---

> > ### Comment · Area_Chair_xgxF · 2025-11-26
> >
> > Hi authors. My interpretation of Reviewer h1Zj’s ethics comment is that it concerns whether your study complied with your institution’s IRB policies. For example, the paper now states:
> > > Participants were randomly chosen from a departmental computer science forum at an R1 research institution
> >
> > Such institutions typically have policies and procedures for conducting studies with human participants, even for minimal-risk listening tests like this (e.g. exempt determinations). **If you can please confirm that all relevant policies and procedures were followed when conducting this study** then I do not think specialized ethics review is needed. Thanks.

---

> > > ### Author Response · Authors · 2025-11-26
> > > **Response on Ethics Concern**
> > >
> > > We thank the area chair for the thoughtful response. The author team can **confirm** that all relevant institutional polices and procedures were followed in the conducting of our listening study. We hope that this information addresses the aforementioned ethics concerns, and dutifully await the reviewer's response.

---

> > > > ### Comment · Area_Chair_xgxF · 2025-11-26
> > > >
> > > > Thank you for confirming.

---

> ### Comment · Reviewer_h1Zj · 2025-11-27
>
> Thanks for your confirmation. I noticed that the Reviewer UgTQ did not update its review but I believe the novelty concern and experiment explanations have been covered during the reviewing based.
>
> From my point of view, this paper demonstrates a **novel** control method on AR-based music generation model and shows its strengths and limitation with sufficient experiments, which may provide a new solution for improving controllability of this kind of generative methods beyond existing fine-tuning methods. I would be happy to raise my score to 10 to defend for this papers acceptance.

---

### Official Review · Reviewer_UgTQ · 2025-11-01

**Soundness:** 2
**Presentation:** 3
**Contribution:** 1
**Rating:** 2
**Confidence:** 3

**Summary:**

This paper presents a training-free method to steer MusicGen with RFMs. The controllable features include tempo, chords, notes, time signatures etc.

**Strengths:**

1. The idea of fine tuning-free steering of multiple music concepts in a universal way is intriguing, and the time-variant control is a useful direction.

2. In the demo page it seems that some controls (e.g., augmented chords; interval 6) are effective. But it is strange why the probing results are low (see questions 2).

**Weaknesses:**

1. The model focuses on global controls that are described by labels. This is not a novel task and can be done by many fine-tuning methods, using either text-based or other controls. The methodology itself is not novel either.

2. The performance of the model seems to be low. Previous works seem to provide better controllability and musicality as in Wu et al (2024) & Lin et al (2023). There is no comparative experiments against previous fine-tuning methods either.

3. Sec 5.1: For some tasks like notes, chord etc., a pretrained model (e.g., audio chord estimator) provides a better evaluation metrics compared to subjective and objective (by reusing the prober) test.

4. Line 243: incomplete sentence.

**Questions:**

1. Line 257: I cannot quite understand the tempo issue. what happened to the tempo category and what specific methods did you used for tempo?

2. I do not quite understand the correspondence between table 2 and the demo page. From the demo page it seems that the controllability is relatively good but the audio quality/musicality is harmed. However, table 2 shows that the quality is relatively ok but the controllability is low. Why?

3. For time varying controls (changing $\phi(t)$), do you have any demos that can produce i.e., tempo changes or time signature changes within a song? As far as I know these are very difficult for current controllable generation models. Currently, I see no time varying control in the demo page.

---

> ### Author Response · Authors · 2025-11-21
> **Rebuttal #1**
>
> We thank the reviewer for their insightful comments on our draft. Below we address major concerns brought up in the review:
>
> >**The model focuses on global controls that are described by labels. This is not a novel task and can be done by many fine-tuning methods, using either text-based or other controls. The methodology itself is not novel either.**
>
> We thank the reviewer for addressing this, and for giving us the opportunity to clarify where our paper sits in the wider controllable TTM space in terms of novelty. While we understand the reviewers point of view, we’d like to respectively push back on the claims that both the task and methodology are non-novel. The reviewer is correct that global attribute control is, in principle, a well explored task in text-to-music generation. However, outside of features like tempo (in works like Music-ControlNet) or chords (such as in MusTango), there has been little exploration of the breadth of music theoretic controls we investigate here in previous works (such as interval relationships or chord quality). Additionally, the ability to control *global* controls as a *function of time* is, to our knowledge, generally underexplored and thus novel in the present work.
>
> Regarding methodology, while there has been some limited investigation of activation-based interventions in music models (see lines 100-102), none have investigated RFMs as a mechanism for controllability in generative music. This fact, combined with our empirical tuning of RFMs to work for fixed-sampling rate domains like musical audio (through temporal control, layer-wise pruning, and stochastic dropout) are wholly novel contributions.
>
> Based on this discussion, we have updated our draft (specifically, the intro and related works) to make our contribution more explicit given your suggestion.
>
> >**The performance of the model seems to be low. Previous works seem to provide better controllability and musicality as in Wu et al (2024) & Lin et al (2023). There is no comparative experiments against previous fine-tuning methods either.**
>
>
> We thank the reviewer for their comment regarding additional baseline comparisons, as such is of key importance. We’d like to clarify that the stated comparison points are both ***training-based*** methods, compared to our training free method (regarding the base TTM model). While we recognize that the performance of our method is not perfect (and will make this fact more explicitly clear in our limitations section), the performance of our method is *by-construction* bounded by supervised finetuning, especially when using other base models (as done in Wu et al.) or different training sets (Lin et al). As we note in our introduction and related works, the benefit of methods like RFM steering is that it is exceedingly light weight compared to the aforementioned training-based baselines, training in less than a minute on a single GPU, and incurring no slow down during inference (unlike other inference-time methods). Additionally, it is unclear whether the fine-tuning based methods described would even be amenable to data like Syntheory, as both mentioned works require *real* music to fine-tune from, rather than synthetic samples to train the RFMs.
>
> While we were bottlenecked during this rebuttal period in performing a true fine-tuning comparison due to computational constraints, we have added an explicit comparison against an inference-time *prompt-based* baseline (suggested by reviewer 6iX2), where we simply append the control target (such as “Notes: C”) to the prompt into MusicGen.
>
> Across all measurements, prompt-only conditioning provides limited control, rarely above chance (except for notes), and can sometimes even impact quality. MusicRFM has higher accuracy on both external and RFM-based metrics when above a control coefficient of 0.3 on all categories. We also investigated combining such prompt approaches with our RFMs, and found that prompt+RFM steering gives us much higher accuracy, particularly on notes, where prompting does already give a higher baseline than random chance. This last fact highlights a newfound key strength of RFMs, as they are able to work in conjunction with other simple control techniques.

---

> > ### Author Response · Authors · 2025-11-21
> > **Rebuttal #2**
> >
> > >**Sec 5.1: For some tasks like notes, chord etc., a pretrained model (e.g., audio chord estimator) provides a better evaluation metrics compared to subjective and objective (by reusing the prober) test.**
> >
> > This is an important factor that we overlooked in our initial draft, and we sincerely thank the reviewer for suggesting this. In our revision, we now include evaluation metrics based on external pretrained audio analyzers, rather than relying solely on subjective listening or our own RFM probes. Specifically, we added a chromagram-based method for note detection, an Essentia-based chord-quality estimator for chords, and an onset-density estimator for tempo. These metrics operate directly on the audio waveform and therefore provide a RFM probe-free measure of steering accuracy. As shown in the updated results tables (Table 3), these external evaluators confirm the same trends observed with our probes: RFM steering provides substantially stronger controllability than prompt-only conditioning, and the combined prompt+RFM setting performs best overall.
> >
> > To our knowledge, there are no great out-of-the-box evaluation methods for detecting the overall "presence" of the other categories. We believe that the above evaluation on chords, notes, and tempos is solid evidence that RFM steering indeed works well and thus can be extrapolated to the other untested categories.
> >
> > >**Line 243: incomplete sentence.**
> >
> > Thanks for catching this one, we have fixed in our new draft.
> >
> > >**Line 257: I cannot quite understand the tempo issue. what happened to the tempo category and what specific methods did you used for tempo?**
> >
> > Here, we meant that because tempo is our only *regression* target, we report metrics with both positive *and* negative control coefficients (corresponding to faster and slower tempos), instead of just positive ones. In order to have this match with the rest of the table, we combine the generations for metric calculation across each absolute value of the control coefficient. We also show new tempo evaluation metrics using onset-density detection.
> >
> > >**I do not quite understand the correspondence between table 2 and the demo page. From the demo page it seems that the controllability is relatively good but the audio quality/musicality is harmed. However, table 2 shows that the quality is relatively ok but the controllability is low. Why?**
> >
> > We recognize that our initial demo page was limited, and have such updated it with a number of new examples, which should speak a bit more to the breadth of quality and musicality. Regarding Table 2, note that the controllability results here are from the *syntheory probes*. As mentioned in the text, we qualitatively found that despite samples seeming to follow the controls, the overall absolute results from the probes were rather low. We attribute this partly to the distributional mismatch (lines 324-327): these probes, trained on syntheory, may not naturally extrapolate their predictive performance to the generated music. We also think that low predictive accuracy does not necessarily 1-to-1 correlate with perceived “control following” given the global nature of the controls, and plan to investigate this further in future work.

---

### Author Response · Authors · 2025-11-21
**Overall Rebuttal Response**

We thank the reviewers for their insightful comments, and are glad to see that the majority recommend acceptance, highlighting the breadth of our experimental setup (reviewers h1Zj, 6iX2), lightweightness of approach (UgTQ, h1Zj, 6iX2), and overall novelty (h1Zj). Below, we highlight our response to shared useful comments and concerns brought up by the reviewers, specifically the 2 main additions of prompt-conditioning baseline experiments and external evaluations that are not dependent on multiclass RFM-probes.

## Prompt-Based Baseline
First, we add a prompt-only conditioning baseline to directly address concerns about the absence of a simple text-conditioning comparison. For each concept, we append explicit control hints (e.g., “Note: C#”, “Slow tempo”) and evaluate MusicGen without any RFM steering. Across all settings, we find that prompt-only control is consistently outperformed by RFM-based steering. Additionally, we conduct experiments on combining prompt conditioning and RFM steering. We see that for most categories, this experimental setup yields the best accuracy results, and most noticeably for notes, where probe accuracy reaches 95%. By providing these new experimental results, we can concretely show that activation-space steering enables forms of fine-grained control that prompting cannot achieve.

## External Control Accuracy Metrics
Second, we introduce external evaluators that operate directly on the waveform, addressing reviewer concerns regarding reliance on probe-based evaluation. We now report: (i) note accuracy via chromagram energy, (ii) chord accuracy using Essentia’s chord estimator, and (iii) tempo control via onset-based event rates. Overall, we find that RFMs perform much better than simple prompt conditioning, and that combining the two methods yields better results as well. We believe that the correlation between probe accuracy and our external evaluators is solid evidence that RFM steering indeed works well and thus can be extrapolated to the other untested categories if such evaluation methods existed.

Finally, we have clarified all experimental details (dataset usage, prompt sources, sample counts, and evaluation procedures) and corrected the typos within the paper to ensure full reproducibility, addressing the other concerns raised by the reviewers. We have also updated the website with new audio samples (time-controlled, multi-direction) and controls on a variety of prompts.

---

### Meta-Review · Area_Chair_SuHS · 2026-01-05

**Summary:**

The paper proposes MusicRFM, adapting Recursive Feature Machines (RFMs) for training-free, activation-level steering of frozen autoregressive music models (MusicGen). RFM probes trained on synthetic data (Syntheory) extract interpretable "concept directions" for attributes like notes, chords, tempo. During inference, these directions are injected with layer pruning, time-varying schedules, and multi-direction support for dynamic/fine-grained control (similar setting as https://arxiv.org/pdf/2402.14285). Results show improved accuracy (e.g., note control from 0.23 to 0.82) with minimal fidelity loss (CLAP within ~0.02 of baseline), evaluated via probes, external analyzers, and listening tests.

Reviewers praised the lightweight, interpretable approach, breadth of controls (including time-varying/multi-attribute), strong empirical validation (ablations, external metrics, listening test), and clear presentation. Concerns focused on limited novelty (building on RFMs/activation steering), reliance on synthetic data/probes for evaluation, insufficient prompt-based baselines, and unclear advantages over fine-tuning or diffusion controls.

The authors' rebuttal added prompt-conditioning baselines (showing RFMs outperform/complement prompts), external waveform-based evaluators (chromagram, Essentia, onset-density), combined prompt+RFM results, efficiency metrics, and clarified scope (training-free vs. fine-tuning). These strengthened claims of unique benefits and addressed baseline gaps.

**Reviewer Concerns:**

Technical details (baselines, external evaluation, efficiency) were largely resolved. Remaining: perceived incrementality over RFM/steering literature, synthetic data limitations, and broader comparisons (fine-tuning/diffusion).

**Reviewer Scores:**

Initial scores mixed (one 2, one 6, one 8). Rebuttal yielded positive notes, the reviewer with 8 mention strong support; consensus shifted toward acceptance.

---

### Decision · Program_Chairs · 2026-01-26

Accept (Poster)